# SpanDrop: Simple and Effective Counterfactual Learning for Long Sequences

## Abstract

Distilling supervision signal from a long sequence to make predictions is a challenging task in machine learning, especially when not all elements in the input sequence contribute equally to the desired output. In this paper, we propose Span-Drop, a simple and effective data augmentation technique that helps models identify the true supervision signal in a long sequence with very few examples. By directly manipulating the input sequence, SpanDrop randomly ablates parts of the sequence at a time and ask the model to perform the same task to emulate counter-factual learning and achieve input attribution. Based on theoretical analysis of its properties, we also propose a variant of SpanDrop based on the beta-Bernoulli distribution, which yields diverse augmented sequences while providing a learning objective that is more consistent with the original dataset. We demonstrate the effectiveness of SpanDrop on a set of carefully designed toy tasks, as well as various natural language processing tasks that require reasoning over long sequences to arrive at the correct answer, and show that it helps models improve performance both when data is scarce and abundant.

## 1 Introduction

Building effective machine learning systems for long sequences is a challenging and important task, which helps us better understand underlying patterns in naturally occurring sequential data like long texts (Radford et al., 2019), protein sequences (Jumper et al., 2021), financial time series (Bao et al., 2017), etc. Recently, there is growing interest in studying neural network models that can capture long-range correlations in sequential data with high computational, memory, and statistical efficiency, especially widely adopted Transformer models (Vaswani et al., 2017).

Previous work approach long-sequence learning in Transformers largely by introducing computational approaches to replace the attention mechanism with more efficient counterparts. These approaches include limiting the input range over which the attention mechanism is applied (Kitaev et al., 2019) to limiting sequence-level attention to only a handful of positions (Beltagy et al., 2020; Zaheer et al., 2020). Other researchers make use of techniques akin to the kernel trick to eliminate the need to compute or instantiate the costly attention matrix (Peng et al., 2020; Katharopoulos et al., 2020; Choromanski et al., 2020). Essentially, these approaches aim to approximate the original pair-wise interaction with lower cost, and are often interested in still capturing the interactions between every pair of input elements (e.g., the long sequence benchmark proposed by Tay et al., 2020).

In this paper, we instead investigate learning problems for long sequences where not all input elements contribute equally to the desired output. Natural examples that take this form include sentiment classification for long customer review documents (where a few salient sentiment words contribute the most), question answering from a large document (where each question typically requires a small number of supporting sentences to answer), key phrase detection in audio processing (where a small number of recorded frames actually determine the prediction), as well as detecting a specific object from a complex scene (where, similarly, a small amount of pixels determine the outcome), to name a few. In these problems, it is usually counterproductive to try and make direct use of the entire input if the contributing portion is small or sparse, which results in a problem of *underspecification* (i.e., the data does not sufficiently define the goal for statistical models). One approach to address this problem is annotating the segments or neighborhoods that directly contribute to the outcome in the entire input. This could take the form of a subset of sentences that answer a question

or describe the relation between entities in a paragraph (Yang et al., 2018; Yao et al., 2019), which function as explainable evidence that supplements the answer. When such annotation is not feasible, researchers and practitioners often need to resort to either collecting more input-output pairs or designing problem-specific data augmentation techniques to make up for the data gap. For real-valued data, this often translates to random transformations (*e.g.*, shifting or flipping an image); for symbolic data like natural language, techniques like masking or substitution are more commonly used (*e.g.*, randomly swapping words with a special mask token or other words). While these approaches have proven effective in some tasks, each has limitations that prevents it from being well-suited for the underspecification scenario. For instance, while global feature transformations enhance group-invariance in learned representations, they do not directly help with better locating the underlying true stimulus. On the other hand, while replacement techniques like masking and substitution help ablate parts of the input, they are susceptible to the position bias of where the true stimulus might occur in the input. Furthermore, while substitution techniques can help create challenging contrastive examples, it is significantly more difficult to design them for complex symbolic sequences (*e.g.*, replacing a phrase naturally in a sentence).

To address these challenges, we propose SPANDROP, a simple and effective technique that helps models distill sparse supervision signal from long sequences when the problem is underspecified. Similar to replacement-based techniques such as masking and substitution, SPANDROP directly ablates parts of the input at random to construct counterfactual examples that preserve the original supervision signal with high probability. Instead of preserving the original sequence positions, however, SPANDROP directly removes ablated elements from the input to mitigate any bias that is related to the absolute positions of elements (rather than the relative positions between them) in the input. Upon closer examination of its theoretical and empirical properties, we further propose a more effective variant of SPANDROP based on the Beta-Bernoulli distribution that enhances the consistency of the augmented objective function with the original one. We demonstrate via carefully designed toy experiments that SPANDROP not only helps models achieve up to $20\times$ sample-efficiency in low-data settings, but also further reduces overfitting even when training data is abundant. We find that it is very effective at mitigating position bias compared to replacement-based counterfactual approaches, and enhances out-of-distribution generalization effectively. We further experiments on four natural language processing tasks that require models to answer question or extract entity relations from long texts, and demonstrate that SPANDROP can improve the performance of already competitive neural models without any change in model architecture.

## 2 METHOD

In this section, we first formulate the problem of sequence inference, where the model takes sequential data as input to make predictions. Then, we introduce SPANDROP, a simple and effective data augmentation technique for long sequence inference, and analyze its theoretical properties.

### 2.1 PROBLEM DEFINITION

**Sequence Inference**. We consider a task where a model takes a sequence $S$ as input and predicts the output $y$. We assume that $S$ consists of $n$ disjoint but contiguous *spans*, each representing a part of the sequence in order $S = (s_1, \ldots, s_n)$. One example of sequence inference is sentiment classification from a paragraph of text, where $S$ is the paragraph and $y$ the desired sentiment label. Spans could be words, phrases, sentences, or a mixture of these in the paragraph. Another example is time series prediction, where $S$ is historical data, $y$ is the value at the next time step.

**Supporting facts**. Given an input-output pair $(S, y)$ for sequence prediction, we assume that $y$ is truly determined by only a subset of spans in $S$. More formally, we assume that there is a subset of spans $S_{\text{sup}} \subset \{s_1, s_2, \ldots, s_n\}$ such that $y$ is independent of $s_i$, if $s_i \notin S_{\text{sup}}$. In sentiment classification, $S_{\text{sup}}$ could consist of important sentiment words or conjunctions (like "good", "bad", "but"); in time series prediction, it could reflect the most recent time steps as well as those a few cycles away if the series is periodic. For simplicity, we will denote the size of this set $m = |S_{\text{sup}}|$, and restrict our attention to tasks where $m \ll n$, such as those described in the previous section.

### 2.2 SPANDROP

In a long sequence inference task with sparse support facts ($m \ll n$), most of the spans in the input sequence will not contribute to the prediction of $y$, but they will introduce spurious correlation in a low-data scenario. SPANDROP generates new data instances $(\tilde{S}, y)$ by ablating these spans at

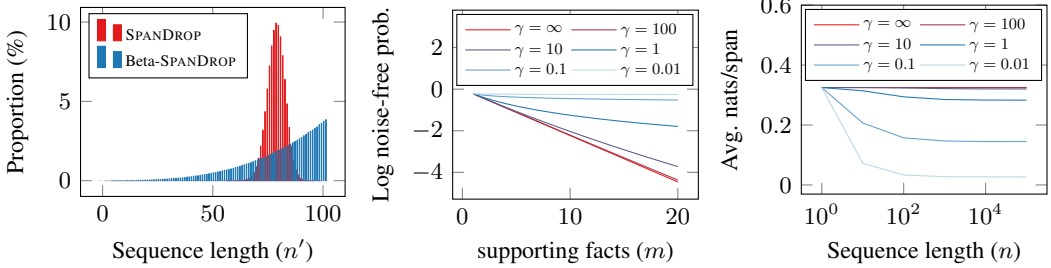

(a) Length of $\tilde{S}$ ($n = 100, p = 0.2$)  (b) supporting fact noise ($p = 0.2$)  (c) Typical set size ($p = 0.1$)

Figure 1: Theoretical comparison between SPANDROP and Beta-SPANDROP.

random, while preserving the supporting facts with high probability so that the model is still trained to make the correct prediction $y$. This is akin to counterfactually determining whether each span truly determines the outcome $y$ by asking what the prediction would have been without it.

**Definition 1** (SPANDROP). *Formally, given a sequence $S$ that consists of spans $(s_1, s_2, \cdots s_n)$, SPANDROP generates a new sequence $\tilde{S}$ as follows:*

$$\delta_i \overset{i.i.d.}{\sim} \mathrm{Bernoulli}(1 - p), \qquad\qquad \tilde{S} = (s_i)_{i=1, \delta_i=1}^n, \qquad (1)$$

*where $p$ is the hyperparameter that determines the probability to drop a span.*

Note that SPANDROP does not require introducing substitute spans or artificial symbols when ablating spans from the input sequence. It makes the most of the natural sequence as it occurs in the original training data, and preserves the relative order between spans that are not dropped, which is often helpful in understanding sequential data (e.g., time series or text). It is also not difficult to establish that the resulting sequence $\tilde{S}$ can preserve all of the $m$ supporting facts with high probability regardless of how large $n$ is.

**Remark 1.** *The new sequence length $n' = |\tilde{S}|$ and the number of preserved supporting facts $m' = |\tilde{S} \cap S_{\mathrm{sup}}|$ follow binomial distributions with parameters $(n, 1 - p)$ and $(m, 1 - p)$, respectively:*

$$P(n'|n, p) = \binom{n}{n'}(1 - p)^{n'} p^{n - n'}, \qquad P(m'|m, p) = \binom{m}{m'}(1 - p)^{m'} p^{m - m'}. \qquad (2)$$

Therefore, the proportion of sequences where all supporting facts are retained (*i.e.*, $m' = m$) is $(1 - p)^m$, which is independent of $n$. This means that as long as the total number of supporting facts in the sequence is bounded, then regardless of the sequence length, we can always choose $p$ carefully such that we end up with many valid new examples with bounded noise introduced to supporting facts. Note that our analysis so far relies only on the assumption that $m$ is known or can be estimated, and thus it can be applied to tasks where the precise set of supporting facts $S_{\mathrm{sup}}$ is unknown. More formally, the amount of new examples can be characterized by the size of the *typical set* of $\tilde{S}$, *i.e.*the set of sequences that the randomly ablated sequence will fall into with high probability. The size of the typical set for SPANDROP is approximately $2^{nH(p)}$, where $H(p)$ is the binary entropy of a Bernoulli random variable with probability $p$. Intuitively, these results indicate that the amount of total counterfactual examples generated by SPANDROP scales exponentially in $n$, but the level of supporting fact noise can be bounded as long as $m$ is small.

However, this formulation of SPANDROP does have a notable drawback that could potentially hinder its efficacy. Because the new sequence length $n'$ follows a binomial distribution, its mean is $n(1 - p)$ and its variance is $np(1 - p)$. For sufficiently large $n$, most of the resulting $\tilde{S}$ will have lengths that concentrate around the mean with a width of $O(\sqrt{n})$, which creates an artificial and permanent distribution drift from the original length (see Figure 1(a)). Furthermore, if we know the identity of $S_{\mathrm{sup}}$ and keep these spans during training, this length reduction will bias the training set towards easier examples to locate spans in $S_{\mathrm{sup}}$. In the next subsection, we will introduce a variant of SPANDROP based on the beta-Bernoulli distribution that alleviates this issue.

### 2.3 BETA-SPANDROP

To address the problem of distribution drift with SPANDROP, we introduce a variant that is based on the beta-Bernoulli distribution. The main idea is that instead of dropping each span in a sequence independently with a fixed probability $p$, we first sample a sequence-level probability $\pi$ at which spans are dropped from a Beta distribution, then use this probability to perform SPANDROP.

**Definition 2** (Beta-SPANDROP). *Let $\alpha = \gamma, \beta = \gamma \cdot \frac{1-p}{p}$, where $\gamma > 0$ is a scaling hyperparameter. Beta-SPANDROP generates $\tilde{S}$ over $S$ as:*

$$\pi \sim \mathrm{B}(\alpha, \beta), \qquad \delta_i \overset{i.i.d.}{\sim} \mathrm{Bernoulli}(1 - \pi), \qquad \tilde{S} = (s_i)_{i=1, \delta_i=1}^n, \qquad (3)$$

*where $\mathrm{B}(\alpha, \beta)$ is the beta-distribution with parameters $\alpha$ and $\beta$.*

It can be easily demonstrated that in Beta-SPANDROP, the probability that each span is dropped is still controlled by $p$, same as in SPANDROP: $\mathbb{E}[\delta_i|p] = \mathbb{E}[\mathbb{E}[\delta_i|\pi]|p] = \mathbb{E}[1 - \pi|p] = 1 - \frac{\alpha}{\alpha+\beta} = 1 - p$. In fact, we can show that as $\gamma \to \infty$, Beta-SPANDROP degenerates into SPANDROP since the beta-distribution would assign all probability mass on $\pi = p$. Despite the simplicity in its implementation, Beta-SPANDROP is significantly less likely to introduce unwanted data distribution drift, while is capable of generating diverse counterfactual examples to regularize the training of sequence inference models. This is due to the following properties:

**Remark 2.** *The new sequence length $n' = |\tilde{S}|$ and the number of preserved supporting facts $m' = |\tilde{S} \cap S_{\mathrm{sup}}|$ follow binomial distributions with parameters $(n, \beta, \alpha)$ and $(m, \beta, \alpha)$, respectively:*

$$P(n'|n, \alpha, \beta) = \frac{\Gamma(n+1)}{\Gamma(n'+1)\Gamma(n-n'+1)} \frac{\Gamma(n'+\beta)\Gamma(n-n'+\alpha)}{\Gamma(n+\alpha+\beta)} \frac{\Gamma(\alpha+\beta)}{\Gamma(\alpha)\Gamma(\beta)}, \qquad (4)$$

$$P(m'|m, \alpha, \beta) = \frac{\Gamma(m+1)}{\Gamma(m'+1)\Gamma(m-m'+1)} \frac{\Gamma(m'+\beta)\Gamma(m-m'+\alpha)}{\Gamma(m+\alpha+\beta)} \frac{\Gamma(\alpha+\beta)}{\Gamma(\alpha)\Gamma(\beta)}, \qquad (5)$$

*where $\Gamma(z) = \int_0^\infty x^{z-1}e^{-x}dx$ is the gamma function.*

As a result, we can show that the probability that Beta-SPANDROP preserves the entire original sequence with the following probability

$$P(n' = n|n, \alpha, \beta) = \frac{\Gamma(n+\beta)\Gamma(\alpha+\beta)}{\Gamma(n+\alpha+\beta)\Gamma(\beta)}. \qquad (6)$$

When $\gamma = 1$, this expression simply reduces to $\frac{\beta}{n+\beta}$; when $\gamma \neq 1$, this quantity tends to $O(n^{-\gamma})$ as $n$ grows sufficiently large. Comparing this to the $O((1-p)^n)$ rate from SPANDROP, we can see that when $n$ is large, Beta-SPANDROP recovers more of the original distribution represented by $(\tilde{S}, y)$ compared to SPANDROP. In fact, as evidenced by Figure 1(a), the counterfactual sequences generated by Beta-SPANDROP are also more spread-out in their length distribution besides covering the original length $n$ with significantly higher probability. A similar analysis can be performed by substituting $n$ and $n'$ with $m$ and $m'$, where we can conclude that as $m$ grows, Beta-SPANDROP is much better at generating counterfactual sequences that preserve the entire supporting fact set $S_{\mathrm{sup}}$. This is shown in Figure 1(b), where the proportion of "noise-free" examples (*i.e.*, $m' = m$) decays exponentially with SPANDROP ($\gamma = \infty$) while remaining much higher when $\gamma$ is sufficiently small. For instance, when $p = 0.1$, $\gamma = 1$ and $m = 10$, the proportion of noise-free examples for SPANDROP is just 34.9%, while that for Beta-SPANDROP is 47.4%.

As we have seen, Beta-SPANDROP is significantly better than its Bernoulli counterpart at assigning probability mass to the original data as well as generated sequences that contain the entire set of supporting facts. A natural question is, does this come at the cost of diverse counterfactual examples? To answer this question we study the entropy of the distribution that $\tilde{S}$ follows by varying $\gamma$ and $n$, and normalize it by $n$ to study the size of typical set of this distribution. As can be seen in Figure 1(c), as long as $\gamma$ is large enough, the average entropy per span $\bar{H}$ degrades very little from the theoretical maximum, which is $H(p)$, attained when $\gamma = \infty$. Therefore, to balance between introducing noise in the supporting facts and generating diverse examples, we set $\gamma = 1$ in our experiments.

**Using the beta-Bernoulli distribution in dropout**. The beta-Bernoulli distribution has been studied in prior work in seeking replacements for the (Bernoulli) dropout mechanism (Srivastava et al.,

2014). Liu et al. (2019a) set $\alpha = \beta$ for the beta distribution in their formulation, which limits the dropout rate to always be 0.5. Lee et al. (2018) fix $\beta = 1$ and vary $\alpha$ to control the sparsity of the result of dropout, which is similar to Beta-SPANDROP when $\gamma = 1$. However, we note that these approaches (as with dropout) are focused more on adding noise to internal representations of neural networks to introduce regularization, while SPANDROP operates directly on the input to ablate different components therein, and thus orthogonal (and potentially complementary) to these approaches. Further, SPANDROP has the benefit of not having to make any assumptions about the model or any changes to it during training, which makes it much more widely applicable.

## 3    FINDCATS: DISTILLING SUPERVISION FROM LONG-SEQUENCES

In this section, we design a synthetic task of finding the animal name "cat" in a character sequence to a) demonstrate the effectiveness of SPANDROP and Beta-SPANDROP in promoting the performance over a series of problems with different settings, b) analyze the various factors that may affect the efficacy of these approaches, and c) compare it to other counterfactual augmentation techniques like masking on mitigating position bias.

### 3.1    EXPERIMENTAL SETUP

**FINDCATS**. To understand the effectiveness of SPANDROP and Beta-SPANDROP in an experimental setting, we designed a synthetic task called FINDCATS where the model is trained to discern that given an animal name "cat", whether a character string contains it as a subsequence (*i.e.*, contains characters in "cat" in order, for instance, "ab**c**d**a**fgbijk**t**ma") or not (*e.g.*, "**abc**defh**t**ijklmn"). This allows us to easily control the total sequence length $n$, the supporting facts size $m$, as well as easily estimate the supporting fact noise that each SPANDROP variant might introduce. To generate the synthetic training data of FINDCATS, we first generate a sequence consisting of lowercase letters (a to z) that does not contain "cat" as a subsequence. For half of these sequences, we label the tuple (cat, $S$) with a negative class to indicate that $S$ does not contain "cat" as a subsequence; for the other half, we choose arbitrary (but not necessarily contiguous) positions in $S$ to replace the letters with letters in "cat" from left to right to generate positive examples.

In all of our experiments, we evaluate model performance on a held-out set of 10,000 examples to observe classification error. We set sequence length to $n = 300$ where each letter is a separate span, and chose positions for the letters in the animal name "cat" uniformly at random in the sequence unless otherwise mentioned.

**Model**. We employ three-layer Transformer model (Vaswani et al., 2017) with position embeddings (Devlin et al., 2019) as the sequence encoder, which is implemented with HuggingFace Transformers (Wolf et al., 2019). For each example ("cat", $S$, $y$), we feed "[CLS] cat [SEP] $S$ [SEP]" to the sequence encoder and then construct binary classifier over the output representation of "[CLS]" to predict $y$. To investigate the effectiveness of SPANDROP, we simply apply SPANDROP to $S$ first before feeding the resulting sequence into the Transformer classifier.

### 3.2    RESULTS AND ANALYSIS

In each experiment, we compare SPANDROP and Beta-SPANDROP at the same drop ratio $p$. And we further use rejection sampling to remove examples that do not preserve the desired supporting facts to understand the effect of supporting fact noise.

**Data efficiency**. We begin by analyzing the contribution of SPANDROP and Beta-SPANDROP to improving the sample efficiency of the baseline model. To achieve this goal, we vary the size of the training set from 10 to 50,000 and observe the prediction error on the held-out set. We observe from the results in Figure 2(a) that: 1) Both SPANDROP and Beta-SPANDROP significantly improve data efficiency in low-data settings. For instance, when trained on only 200 training examples, SPANDROP variants can achieve the generalization performance of the baseline model trained on 5x to even 20x data. 2) Removing supporting fact noise typically improves data efficiency further by about 2x. This indicates it is helpful not to drop spans in $S_{\text{sup}}$ during training when possible, so that the model is always trained with true counterfactual examples rather than sometimes noisy ones. 3) Beta-SPANDROP consistently improves upon the baseline model even when data is abundant. This

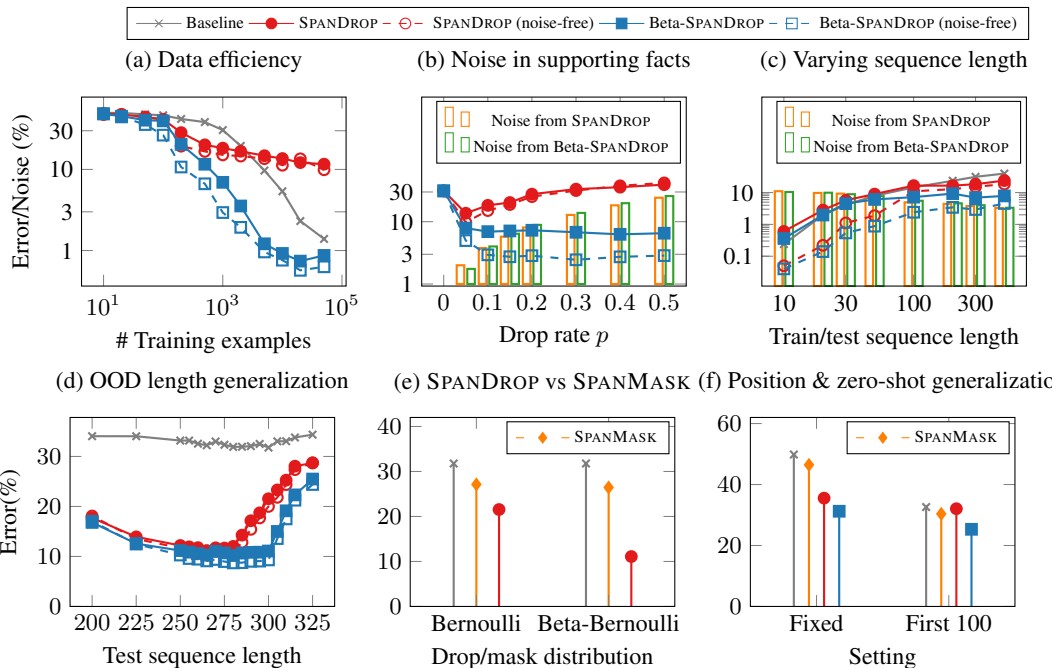

Figure 2: Experimental results of SPANDROP variants and SPANMASK on the FINDCATS synthetic tasks.

is likely due to the difficulty of the task when $n = 300$ and $m = 3$. Similar to many real-world tasks, the task remains underspecified even when the generalization error is already very low thanks to the large amount of training data available. 4) SPANDROP introduces inconsistent training objective with the original training set, which leads to performance deterioration when there is sufficient training data, which is consistent with our theoretical observation.

**Effect of supporting fact noise and sequence length.** Since SPANDROP introduces noise in the supporting facts (albeit with a low probability), it is natural to ask if such noise is negatively correlated with model performance. We study this by varying the drop ratio $p$ from $\{0.05, 0.1, 0.15, 0.2, 0.3, 0.4, 0.5\}$ on fixed training sets of size 1,000, and observe the resulting model performance and supporting fact error. As can be seen in Figure 2(b), supporting fact noise increases rapidly as $p$ grows.[1] However, we note that although the performance of SPANDROP deteriorates as $p$ increases, that of Beta-SPANDROP stays relatively stable. Inspecting these results more closely, we find that even the performance of the noise-free variants follow a similar trend, which should not be affected by supporting fact noise.

Recalling the observations from our data efficiency experiments, we next turn to the hypothesis that this discrepancy is mainly caused by the inconsistent length distribution SPANDROP introduces. To test this hypothesis, we conduct two separate sets of experiments: 1) training and testing the model on varying sequence lengths $\{10, 20, 30, 50, 100, 200, 300, 500\}$, where longer sequences suffer more from the discrepancy between SPANDROP-resulted sequence lengths and the original sequence length; and 2) testing the model trained on $n = 300$ on test sets of different lengths, and if our hypothesis about distribution drift were correct, we should see SPANDROP models' performance peaking around $n' = n(1-p)$, while the performance of Beta-SPANDROP is less affected by sequence length. As can be seen from Figures 2(c) and 2(d), our experimental results seem to well supporting this hypothesis. Specifically, in Figure 2(c), while the performance of both SPANDROP variants deteriorates as $n$ grows and the task becomes more challenging and underspecified, SPAN-DROP deteriorates at a faster speed even when we remove the effect of supporting fact noise. On the other hand, we can clearly see in Figure 2(d) that SPANDROP performance peak around sequences of length 270 ($= n(1-p) = 300 \times (1-0.1)$) before rapidly deteriorating, while Beta-SPANDROP is unaffected until test sequence length exceeds that of all examples seen during training.

---

[1]Note that the noise in our experiments are lower than what would be predicted by theory, because in practice the initial sequence $S$ might already contain parts of "cat" before it is inserted. This creates redundant sets of supporting facts for this task and reduces supporting fact noise especially when $n$ is large.

**Mitigating position bias**. Besides SPANDROP, replacement-based techniques like masking can also be applied to introduce counterfactual examples into sequence model training, where elements in the sequence are replaced by a special symbol that is not used at test time. We implement SPANMASK in the same way as SPANDROP except spans are replaced rather than removed when the sampled "drop mask" $\delta_i$ is 0. We first inspect whether SPANMASK benefits from the same beta-Bernoulli distribution we use in SPANDROP. As can be seen in Figure 2(e), the gain from switching to a beta-Bernoulli distribution provides negligible benefit to SPANMASK, which does not alter the sequence length of the input to begin with. We also see that SPANMASK results in significantly higher error than both SPANDROP and Beta-SPANDROP in this setting. We further experiment with introducing position bias into the training data (but not the test data) to test whether these method help the model generalize to an unseen setting. Specifically, instead of selecting the position for the characters "cat" uniformly at random, we train the model with a "fixed position" dataset where they always occur at indices (10, 110, 210), and a "first 100" dataset where they are uniformly distributed among the first 100 letters. As can be seen in Figure 2(f), both the baseline and SPANMASK models overfit to the position bias in the "fixed" setting, while SPANDROP techniques significantly reduce zero-shot generalization error. In the "first 100" setting, Beta-SPANDROP consistently outperforms its Bernoulli counterpart and SPANMASK at improving the performance of the baseline model as well, indicating that SPANDROP variants are effective at reducing the position bias of the model.

## 4    EXPERIMENTS ON NATURAL LANGUAGE DATA

To examine the efficacy of the proposed SPANDROP techniques on realistic data, we conduct experiments on four natural language processing datasets that represent the tasks of single- and multi-hop extractive question answering, multiple-choice question answering, and relation extraction. We focus on showing the effect of SPANDROP instead of pursuing the state of the art in these experiments.

**Datasets**. We use four natural language processing datasets: **SQuAD** 1.1 (Rajpurkar et al., 2016), where models answer questions on a paragraph of text from Wikipedia; **MultiRC** (Khashabi et al., 2018), which is a multi-choice reading comprehension task in which questions can only be answered by taking into account information from *multiple sentences*; **HotpotQA** (Yang et al., 2018), which requires models to perform multi-hop reasoning over multiple Wikipedia pages to answer questions; and **DocRED** (Yao et al., 2019), which is a *document-level* data set for relation extraction.

For the SQuAD dataset, we define spans as collections of one or more consecutive tokens to show that SPANDROP can be applied to different granularities. For the rest three datasets, we define spans to be sentences since supporting facts are provided at sentence level. For all of these tasks, we report standard exact match (EM) and $F_1$ metrics where applicable, for which higher scores are better. We refer the reader to the appendix for details about the statistics and metrics of these datasets.

**Model**. We build our models for these tasks using ELECTRA (Clark et al., 2019), since it is shown to perform well across a range of NLP tasks recently. We introduce randomly initialized task-specific parameters designed for each task following prior work on each dataset, and finetune these models on each dataset to report results. We refer the reader to the appendix for training details and hyperparameter settings.

**Main results**. We first present the performance of our implemented models and their combination with SPANDROP variants on the four natural language processing tasks. We also include results from representative prior work on each dataset for reference (detailed in the appendix), and summarize the results in Table 1. We observe that: 1) our implemented models achieve competitive and sometimes significantly better performance (in the cases of HotpotQA, SQuAD, and DocRED) compared to published results, especially considering that we do not tailor our models to each task too much; 2) SPANDROP improves the performance over these models even when the training set is large and that the model is already performing well; 3) Models trained with Beta-SPANDROP consistently perform better or equally well with their SPANDROP counterparts across all datasets, demonstrating that our observations on the synthetic datasets generalize well to real-world ones. We note that the performance gains on real-world data is less significant, which likely results from the fact spans in the synthetic task are independent from each other, which is not the case in natural language data.

We further evaluate the performance of our trained models on the MultiRC testing data, and obtain results of EM/F1: 41.1/79.8, 39.9/78.5 and 39.1/78.2 for models with Beta-SPANDROP, SPANDROP,

(a) HotpotQA dev

| Model | Ans $F_1$ | Sup $F_1$ | Joint $F_1$ |
|---|---|---|---|
| RoBERTa-base | 73.5 | 83.4 | 63.5 |
| Longformer-base | 74.3 | 84.4 | 64.4 |
| SAE BERT-base | 73.6 | 84.6 | 65.0 |
| *Our implementation* | | | |
| ELECTRA-base | 74.2 | 86.3 | 66.2 |
| + SPANDROP | 74.7 | 86.7 | 66.8 |
| + Beta-SPANDROP | 74.7 | 86.9 | 67.1 |

(b) MultiRC dev

| Model | EM | $F_1$ |
|---|---|---|
| BERT-base | 26.6 | 74.2 |
| RoBERTa-base | 38.7 | 79.2 |
| REPT RoBERTa-base | 40.4 | 80.0 |
| *Our implementation* | | |
| ELECTRA-base | 40.1 | 80.4 |
| + SPANDROP | 42.3 | 81.7 |
| + Beta-SPANDROP | 44.8 | 81.6 |

(c) DocRED dev

| Model | Ign $F_1$ | RE $F_1$ | Evi $F_1$ |
|---|---|---|---|
| E2GRE BERT-base | 55.2 | 58.7 | 47.1 |
| ATLOP BERT-base | 59.2 | 61.1 | — |
| SSAN BERT-base | 57.0 | 59.2 | — |
| *Our implementation* | | | |
| ELECTRA-base | 59.6 | 61.6 | 50.8 |
| + SPANDROP | 59.9 | 61.9 | 51.2 |
| + Beta-SPANDROP | 60.1 | 62.1 | 51.2 |

(d) SQuAD dev

| Model | EM | $F_1$ |
|---|---|---|
| RoBERTa-base | — | 90.6 |
| ELECTRA-base | 84.5 | 90.8 |
| XLNet-large | 89.7 | 95.1 |
| *Our implementation* | | |
| ELECTRA-base | 86.6 | 92.4 |
| + SPANDROP | 86.8 | 92.6 |
| + Beta-SPANDROP | 86.9 | 92.7 |

Table 1: Main results on four natural language processing datasets.

and without SPANDROP, respectively. This indicates that both Beta-SPANDROP and SPANDROP improve the model generalization ability, and Beta-SPANDROP is better than SPANDROP, improving EM/F1 with 2.0/1.6 absolute over the baseline.

Next, to better understand whether the properties of SPANDROP and Beta-SPANDROP we observe on the synthetic data generalize to real-world data, we further perform a set of analysis experiments on SQuAD. Specifically, we are interested in studying the effect of the amount of training data, the span drop ratio $p$, and the choice of span size on performance.

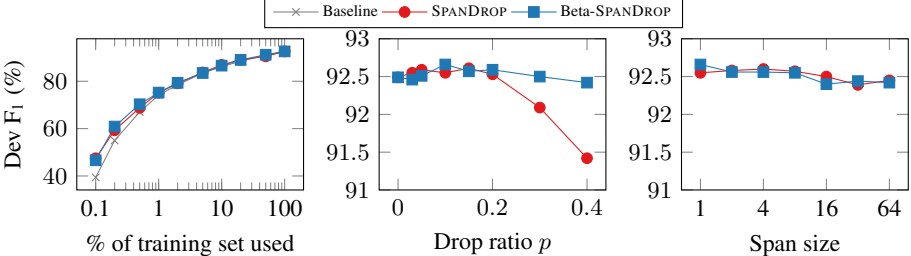

Figure 3: Effect analysis of training data size, drop ratio and span size on performance of models trained with SPANDROP and Beta-SPANDROP over SQuAD

**Effect of low data**. To understand SPANDROP's regularizing effect when training data is scarce, we study the model's generalization performance when training on only 0.1% of the training data (around 100 examples) to using the entire training set (around 88k examples). As can be seen in Figure 3 (left), both SPANDROP and Beta-SPANDROP significantly improve model performance when the amount of training data is extremely low. As the amount of training data increases, this gap slowly closes but remains consistently positive. The final gap when 100% of the training data is used is still sufficient to separate top-2 performing systems on this dataset.

**Impact of drop ratio**. We compare SPANDROP and Beta-SPANDROP by controlling how likely each span is dropped on average (drop ratio $p$). Recall from our experiments on FINDCATS that larger $p$ will result in distribution drift from the original training set for SPANDROP but not Beta-SPANDROP, thus the performance of the former deteriorates as $p$ increases while the latter is virtually not affected. As can be seen in Figure 3 (middle), our observation on real-world data is consistent with this theoretical prediction, and indicate that Beta-SPANDROP is a better technique for data augmentation should one want to increase sequence diversity by setting $p$ to a larger value.

**Impact of span size**. We train the model with SPANDROP on SQuAD with varying span sizes of $\{1, 2, 4, 8, 16, 32, 64\}$ tokens per span to understand the effect of this hyperparameter. We observe in Figure 3 (right) that as span size grows, the generalization performance of the model first holds roughly constant, then slowly deteriorates as span size grows too large. This suggests that the main contributors to generalization performance might have been the total number of spans in the entire sequence, which reduces with larger spans. This results in fewer potential augmented sequences for counterfactual learning, therefore lowering regularization strength. This observation is consistent with that on our synthetic data in our preliminary experiments, where we see that controlling for other factors, larger span sizes yield deteriorated generalization performance (data not shown due to space limit). This also suggests that while SPANDROP works with arbitrary span sizes, the optimal choice of spans for different tasks warrants further investigation, which we leave to future work.

## 5    RELATED WORK

**Long Sequence Inference**. Many applications require the prediction/inference over long sequences, such as multi-hop reading comprehension (Yang et al., 2018; Welbl et al., 2018), long document summarization (Huang et al., 2021), document-level information extraction (Yao et al., 2019) in natural language processing, long sequence time-series prediction (Zhou et al., 2021a), promoter region and chromatin-profile prediction in DNA sequence (Oubounyt et al., 2019; Zhou & Troyanskaya, 2015) in Genomics etc, where not all elements in the long sequence contribute equally to the desired output. Aside from approaches we have discussed that attempt to approximate all pair-wise interactions between elements in a sequence, more recent work has also investigated compressing long sequences into shorter ones to distill the information therein for prediction or representation learning (Rae et al., 2020; Goyal et al., 2020; Kim & Cho, 2021).

**Sequence Data Augmentation**. Data augmentation is an effective common technique for under-specified tasks like long sequence inference. Feng et al. (2021) propose to group common data augmentation techniques in natural language processing into three categories: 1) rule-based methods (Zhang et al., 2015; Wei & Zou, 2019; Şahin & Steedman, 2018), which apply a set of pre-defined operations over the raw input, such as removing, adding, shuffling and replacement; 2) example mixup-based methods (Guo et al., 2019; Guo, 2020; Chen et al., 2020; Jindal et al., 2020), which, inspired from Mixup in computer vision (Zhang et al., 2018), perform interpolation between continuous features like word embeddings and sentence embeddings; 3) model-based methods (Xie et al., 2020; Sennrich et al., 2016), which use trained models to generate new examples (*e.g.*, back translation Xie et al., 2020).

Most of existing rule-based data augmentation methods operate at the token/word level (Feng et al., 2021), such as word shuffle/replacement/addition (Wei & Zou, 2019). Shuffle-based techniques are less applicable when order information is crucial in the raw data (Lan et al., 2019, *e.g.*, in natural language). Moreover, these operations might not be trivial in implementation over larger spans (*e.g.*, at the phrase or sentence level). For example, while replacing tokens require selecting candidates from a fixed vocabulary which can be provided by well estimated language models (Clark et al., 2019), replacing phrases or sentences is significantly more challenging since the "vocabulary" is unbounded and marginal probability difficult to estimate. In contrast, our proposed SPANDROP supports data augmentation in multiple granularity as the spans in SPANDROP can be of any length, and is able to reserve sequence order since drop operation does not change the relative order of the original input.

## 6    CONCLUSION

In this paper, we presented SPANDROP, a simple and effective method for learning from long sequences, which ablates parts of the sequence at random to generate counterfactual data to distill the sparse supervision signal that is predictive of the desired output. We show via theoretical analysis and carefully designed synthetic datasets that SPANDROP and its variant based on the beta-Bernoulli distribution help model achieve competitive performance with a fraction of the data by introducing diverse augmented training examples, and generalize better to previously unseen data. Our experiments on four real-world NLP datasets demonstrate that besides these benefits, SPANDROP can further improve upon powerful pretrained Transformer models even when data is abundant.

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
