# OpenReview forum: "SpanDrop: Simple and Effective Counterfactual Learning for Long Sequences"
_ICLR.cc/2022/Conference — ICLR 2022 Submitted_

### Official Review · Reviewer_ykje · 2021-10-30

**Correctness:** 3
**Technical Novelty And Significance:** 2
**Empirical Novelty And Significance:** 2
**Recommendation:** 3
**Confidence:** 3

**Main Review:**

Strength:

Effective data augmentation is a relevant topic.

The paper is well written.

Weakness:

Limited novelty.

My major concern is the lack of comparison of other baselines. In section5, the author discuss a bunch of related works on data augmentation, however, in natural language experiments these are not compared. I know that SCANDROP is easier to implement, but that does not mean you don't need to compare with other algorithms.

It's not surprising that this data augmentation is effective, especially for the catfinding task.

While the improvement over electra-base is consistent, it's not large.

Below are detailed comments:

Sec1. I don't quite understand why when in data the contributing portion is small or sparse, then the problem is "underspecified". It's still specified by the small contributing portion I think? Are you suggesting the contributing part is so small, that for example, a classifier can not reach a decision?

What not span-replace? If you replace a span with a random token, it gives counterfactual sequence with the same length.
Minor comments:

Sec1. "the data does not sufficiently define the goal for statistical models", I don't understand what global is here.


**Summary Of The Paper:**

In this paper, we instead investigate learning problems for long sequences where not all input elements contribute equally to the desired output. SCANDROP is a simple algorithm to randomly drop segments in a sequence. The authors first establish that when the number of contributing segments is sparse, the algorithm will preserve them with a relatively large probability. Then, Beta-SCANDROP is proposed to preserve the original sequence length with higher probability. In experiments, consistent improvement is shown.


**Summary Of The Review:**

I have two major concerns (1) limited novelty. (2) The lack of comparison of other baselines. In section5, the author discuss a bunch of related works on data augmentation, however, in natural language experiments these are not compared. I know that SCANDROP is easier to implement, but that does not mean you don't need to compare with other algorithms.

---

> ### Author Response · Authors · 2021-11-12
> **Thank you for the constructive comments.**
>
> We thank the reviewer for the constructive comments.
>
> Q1:  limited novelty
>
> R1: Please refer to the general response to all reviewers for our explanation about the novelty of our work, including not only SpanDrop, a new data augmentation technique, but also the mathematical formulation of the problem of long sequence with sparse supporting facts, based on which we also present Beta-SpanDrop, a variant of SpanDrop that is based on the Beta-Bernoulli distribution.
>
> Q2: lack of comparison of other baselines. What not span-replace? If you replace a span with a random token, it gives counterfactual sequence with the same length.
>
> R2: In our experiments on synthetic data, we have compared SpanDrop against SpanMask, which replaces a span with a fixed artificial token. We find that despite the fact that SpanMask can generate counterfactual sequences with the same length as the original sequence, it actually leads to overfitting in certain cases when there is strong spurious position bias for supporting facts in the training data (Figure 2(f)). Furthermore, without altering the input length, SpanMask does not enjoy the added variance in input patterns introduced by our Beta-Bernoulli modification (Figure 2(e)).
>
> For real-world datasets, we considered spans of various sizes (words, contiguous words, and sentences), which is sometimes informed by the underlying task. In cases especially when spans extend beyond a single token, the choice of replacement can be numerous. For instance, the replacement can be a collection of random tokens of the same length, a single random token, a random span/sentence from other contexts, or a span/sentence carefully chosen to construct a natural-looking example. This results in a large number of factors that may determine the performance of these data augmentation techniques, and makes it difficult to conclusively compare them with SpanDrop. In contrast, we have demonstrated that with just two tunable hyperparameters and a simple operation on the input, SpanDrop can improve the performance of competitive baseline models on each of the tasks we experimented with, which we believe demonstrates its greater potential for wide adoption and therefore improvement on many tasks.
>
> Q3:  I don't quite understand why when in data the contributing portion is small or sparse, then the problem is "underspecified". It's still specified by the small contributing portion I think? Are you suggesting the contributing part is so small, that for example, a classifier can not reach a decision?
>
> R3: Here, by “underspecified” we mean that when the amount of supporting facts is small in each training instance, a much larger amount of data from the true distribution would be needed to “statistically specify” that the true input-output mapping in the data is between just the supporting facts and the desired output, or equivalently, rule out the spurious correlation between other input elements in the long sequence and the desired output in the training data. In practice, this means that a statistically inefficient model/classifier would not be able to reach an input-output mapping that is capable of generalizing to unseen data, if the amount of training data is limited. We will try to clarify this point further in the revision of our paper.
>
>
> Q4: Sec1. "the data does not sufficiently define the goal for statistical models", I don't understand what global is here.
>
> R4: Assuming by “global” the reviewer means “goal”, we mean that the amount of data available might not sufficiently establish the input-output mapping where only up to $m$ supporting facts in the input truly determine the output under the ground truth data distribution, if we were to just use these sequence-output pairs as is. In these cases, SpanDrop can help improve the statistical efficiency of a statistical model without making many assumptions about the data, where the $m$ supporting facts are, or what kind of statistical model is applied.

---

> > ### Comment · Reviewer_ykje · 2021-11-30
> > **Thanks!**
> >
> > Thanks for the response!

---

### Official Review · Reviewer_qHWi · 2021-11-02

**Correctness:** 4
**Technical Novelty And Significance:** 2
**Empirical Novelty And Significance:** 1
**Recommendation:** 3
**Confidence:** 4

**Main Review:**

Strengths: 1. Simple procedure to improve accuracy for long sequence input problems. 2. Mathematically sound. The authors prove the claims regarding augmented sequence lengths.

Weaknesses:
1. The procedure is fairly simple, and similar ideas have been used for regularizing models. I find the solution similar to word dropout, and would be interesting to see comparison to that.
2. The gains on real datasets are not high.

**Summary Of The Paper:**

The paper focuses on distilling supervision signal from long sequences. They focus on cases where the input is a long sequence of length n, but the target prediction is determined by a small subset of size m of sequence fragments, where m << n. The authors propose augmenting data by randomly dropping spans from the input sequence. They first propose SpanDrop which removes each span with a probability p. This process might result in shorter sequences, resulting in shift of training data distributions. As a fix, they also propose Beta-SpanDrop where the spans are dropped using probabilities sampled from beta bernoulli distribution. Beta SpanDrop preserves the length of the original utterance with higher probability, while generating similar variety of augmented utterances.

**Summary Of The Review:**

The augmentation procedure is very similar to word dropout. It is not clear if similar benefits can be obtained by using word dropout at training time. The experimental results on real datasets are weak.

---

> ### Author Response · Authors · 2021-11-12
> **Thank you for the helpful feedback.**
>
> We thank the reviewer for the helpful feedback.
>
> Q1: Similar to word dropout, and would be interesting to see comparison to that.
>
> R1: Please refer to the general response to all reviewers for our explanation about the novelty of our work, including not only SpanDrop, a new data augmentation technique and how it is different from commonly used notions of “word dropout”, but also the mathematical formulation of the problem of long sequence with sparse supporting facts, based on which we also present Beta-SpanDrop, a variant of SpanDrop that is based on the Beta-Bernoulli distribution.
>
> Q2: The gains on real datasets are not high.
>
> R2: Though the absolute gain on real datasets appear moderate, the relative performance improvement is still competitive on the corresponding task. For example, on the HotpotQA dataset, Beta-SpanDrop improved the Joint F1 metric by 0.9. To put this number in perspective, this is equal to the improvement achieved by the Longformer model compared to RoBERTa. The RoBERTa-to-Longformer improvement resorts to more elaborate model design and costly pre-training. Similarly, on the MultiRC dataset, Beta-SpanDrop improves the the EM metric by up to 4.7, which is significantly larger than the 1.7 achieved by REPT (a model based on RoBERTa) compared to RoBERTa, when the latter requires retrieval-based pretraining to achieve this gain. In contrast, SpanDrop is relatively simple to implement, and treats the model in question largely as a black-box
>
> Meanwhile, all these improvements are achieved relative to baseline models that we have implemented ourselves, which achieve competitive and sometimes significantly better performance (such as, HotpotQA, SQuAD, and DocRED) compared to published results.

---

### Official Review · Reviewer_DeC3 · 2021-11-02

**Correctness:** 3
**Technical Novelty And Significance:** 2
**Empirical Novelty And Significance:** 2
**Recommendation:** 3
**Confidence:** 4

**Main Review:**

Strengths:
- The proposed method is simple and easy to understand.
- The method outperforms baseline methods.


Weaknesses
- Several technical details are not discussed.
    - How to partition a long sequence into spans? For text sequences? For time series? Will the final results be sensitive to the partition of spans?
    - If each word in NLP sequences is a span, the proposed method is the same as the word-level dropout baseline studied in previous works, e.g., [1]. What's the tech novelty of this work?
    - While the authors focus on the "learning problems for long sequences where not all input elements contribute equally to the desired output", the uniqueness of this kind of problems is not clear to me. In most (if not all) real-world sequence classification and prediction problems, not all input elements contribute equally to the desired output; otherwise, the problems will become much easier.
- Experiments need to be improved and enhanced, e.g.,
    - As reviewed in the second paragraph, there are many Transformer variants proposed for long sequences, but none of them is compared. Although these approaches aim to approximate the original pairwise interaction with lower cost and are often interested in still capturing the interactions between every pair of input elements (e.g., the long sequence benchmark proposed by Tay et al., 2020), they can still be directly applied to long sequences where not all input elements contribute equally to the desired output.
    - The four NLP datasets used in experiments are not representative tasks for long sequences. I suggest to test on the public benchmark datasets, e.g., [2].


[1] Soft Contextual Data Augmentation for Neural Machine Translation, ACL 2019.

[2] Long range arena: A benchmark for efficient transformers. In International Conference on Learning Representations, 2020.

**Summary Of The Paper:**

This work proposes SPANDROP, a simple variant of dropout, working on the spans of long sequences. SPANDROP randomly ablates parts of a sequence at a time and asks the model to perform the same task to emulate counterfactual learning and achieve input attribution. The method is tested on both toy tasks and four NLP tasks.

**Summary Of The Review:**

Both the technical part and empirical evaluations can be improved.

---

> ### Author Response · Authors · 2021-11-12
> **Thank you for the constructive feedback.**
>
> We thank the reviewer for the constructive feedback.
>
> Q1: How to partition a long sequence into spans? For text sequences? For time series? Will the final results be sensitive to the partition of spans?
>
> R1: For our experiments on text sequences, we experimented with spans that are single tokens, single sentences, or the combination of a few consecutive tokens (see the second paragraph of the “Datasets” portion of Section 4). In our experiments, we found a relatively small effect different partitions have on model performance (Figure 3, right, and the “Impact of span size” paragraph on Page 9), which might be the result of our combination of task, model, and hyperparameter selection. For time series or data of other modality, we believe a similar principle could apply, where spans could be a single point in time, or a local neighborhood in time (e.g., a minute, an hour, a day, etc.).
>
> While different partitions yielded similar results in our experiments, we believe further investigation of optimal span selection might be warranted. Previous work has observed non-negligible difference in downstream performance from pretrained BERT models and their whole-word-masking counterparts, which indicates that task-specific selection of partitioning might further help improve the performance of SpanDrop-based models. We leave this investigation to future work.
>
> Q2: What's the tech novelty of this work? Comparison to Word dropout
>
> R2: Please refer to the general response to all reviewers for our explanation about the novelty of our work, including not only SpanDrop, a new data augmentation technique and how it is different from commonly used notions of “word dropout”, but also the mathematical formulation of the problem of long sequence with sparse supporting facts, based on which we also present Beta-SpanDrop, a variant of SpanDrop that is based on the Beta-Bernoulli distribution.
>
> Q3: The uniqueness of this kind of problems is not clear to me. No experiments over long sequence benchmark proposed by Tay et al., 2020. Experiments need to be improved and enhanced
>
> R3: As is also discussed in the third point in our general response to reviewers, we formulate and attack a problem that is focused on inference over long sequences with sparse supporting facts, i.e., only a few spans determine the desired output. This is different from most sequence benchmarks in Tay et al., 2020, where all elements in the sequence can conceivably determine the desired output. Therefore, most of these benchmarks focus on investigating model ability in capturing the whole sequence, rather than teasing out the few elements that directly contribute to the output via statistical training. While both are valid research problems, we believe that the research problem we formulated is rooted in many real-world tasks that rely on statistical learning methods to make predictions.

---

### Official Review · Reviewer_EEvq · 2021-11-03

**Correctness:** 4
**Technical Novelty And Significance:** 3
**Empirical Novelty And Significance:** 3
**Recommendation:** 8
**Confidence:** 5

**Main Review:**

The method is simple yet effective. Experiments in the paper are well-designed to show the effectiveness of the method.

The method generally assumes that the decision of drop can be done independently. However, the positions of salient information would usually exhibit certain patterns such as appearing close to each other. Do you have any idea to incorporate those characteristics into the method or learn those patterns?

As the name of the method is SpanDrop, it would be much interesting if various selection/split methods of spans have been investigated. Of course, splitting by a sentence for passages is natural.

SpanDrop is only applied at the input level. I am curious whether this drop could be applied to intermediate representations like done in LengthDrop (Kim and Cho, 2021).

I wonder about the additional training cost of SpanDrop compared to the standard training. The expectation of length is the same for SpanDrop and Beta-SpanDrop. However, the expectation of length’s square would be different. Therefore, I guess their computational overhead will be also different. The sampling cost would be almost negligible. Does it require additional training steps to converge due to its regularization effect? Don’t it require modification of other regularization such as standard dropout?

Do you have any thoughts about whether SpanDrop will be generalized seamlessly to other tasks than NLP?


**Summary Of The Paper:**

This paper proposes a data augmentation method, SpanDrop (and its variant Beta-SpanDrop), for long sequence data, especially where supporting facts take small portions. They provide a theoretical background on their method and evaluate the method on a synthetic task (FindCats) and four real natural language processing tasks requiring reasoning over long texts. SpanDrop is effectively improves the accuracy in both low-resource and abundant-resource settings.


**Summary Of The Review:**

The paper is well-written and well-structured. The proposed method is theoretically and empirically verified. Also, the paper is practically useful because it is easy to implement and plug into any tasks handling (long) sequences for better accuracy.

---

> ### Author Response · Authors · 2021-11-12
> **Thank you for the positive and detailed feedback.**
>
> We thank the reviewer for the positive and detailed feedback.
>
> Q1: The method generally assumes that the decision of drop can be done independently. However, the positions of salient information would usually exhibit certain patterns such as appearing close to each other. Do you have any idea to incorporate those characteristics into the method or learn those patterns?
>
> R1: This is an interesting question! According to Remarks 1 and 2 in the paper, the probability of dropping salident information (i.e., supporting facts) depends on the number of supporting facts $m$ only, and is independent of the positions of these salient information. Therefore, the pattern of salient information would not necessarily affect the performance of SpanDrop. In our synthetic experiments, we also investigate the effectiveness of our methods by varying the positions of the supporting facts appearing in the sequence, and find SpanDrop-trained models suffer less from spurious correlation with input positions (See the last sub-figure of Fig. 2).
>
> However, if we know in advance that the positions of salident information appear close to each other in our task of interest, we can also consider splitting the sequences with large span size (or task-dependent span sizes) to reduce the number of supporting facts $m$. Smaller $m$ means a lower probability of dropping supporting facts when other hyperparameters are held constant (Remarks 1 and 2), and can allow for higher drop ratio $p$ (thus a stronger regularization effect) while maintaining the same level of noise.
>
> Q2: Whether this drop could be applied to intermediate representations like done in LengthDrop (Kim and Cho, 2021).
>
> R2: We could apply our SpanDrop to intermediate representations like done in LengthDrop, if we view the intermediate representations as a sequence of vectors and define each vector as a span. In this case, SpanDrop is equivalent to LengthDrop.
>
> Moreover, our proposed SpanDrop based on the Beta-Bernoulli distribution shows better theoretical properties and experimental performance than SpanDrop based on Bernoulli distribution. It would be interesting for future work to investigate applying the Beta-Bernoulli distribution to other dropout-related techniques as well, such as LengthDrop.
>
>
>
> Q3: The additional training cost of SpanDrop compared to the standard training.  Does it require additional training steps to converge due to its regularization effect? Don’t it require modification of other regularizations such as standard dropout?
>
> R3: Compared to standard training, the models trained with SpanDrop need more training steps to converge. We will add such convergence analysis in our revision. And in our experiments, we implemented SpanDrop for model training without changing any other regularizations (such as standard dropout) and observed noticeable improvements. We leave to future work to investigate the interaction between SpanDrop and other regularization techniques.
>
>
> Q4: Do you have any thoughts about whether SpanDrop will be generalized seamlessly to other tasks than NLP?
>
> R4: While we do not have experimental results, we believe SpanDrop can generalize to other tasks than those in NLP, such as time series classification and image classification, as long as we can formulate the problem as the mapping from a long sequence of spans to a desired output. For time series, spans could be defined as a single point in time or a small local window (a minute, an hour, a day, etc.); for images, spans could be defined as single pixels or image patches laid out by traversing the image in a specific order (e.g., rastering order).

---

> > ### Comment · Reviewer_EEvq · 2021-11-30
> > **Thanks for the answers**
> >
> > Sorry for the late reply! I still like the contributions of this paper. However, the answers are somewhat unsatisfactory for me, so I want to keep comments regarding the response before the discussion period ends.
> >
> > R-R1: As far as I understand, Remarks 1 and 2 assume whether each token is salient is independent of each other.  My question was about the situation where this assumption does not hold.
> > R-R2: My question was not only whether we could apply this drop to intermediate representations but also whether it could lead to performance gain.
> > R-R3: Could you provide approximate numbers?

---

> > > ### Author Response · Authors · 2021-11-30
> > > **Thank you for the follow-up discussion!**
> > >
> > > Thank you for the insightful follow-up questions and discussion!
> > >
> > > R1': To clarify, Remarks 1 and 2 actually make no assumption about the independence in saliency among spans, but reason about the entire collection of $m$ supporting facts as a whole. That is, we are interested in understanding the probability that we are able to preserve **all** $m$ supporting facts in SpanDrop (and treat any incomplete subset of these facts as insufficient evidence for $y$), assuming that each span is **dropped** independently.
> > >
> > > That being said, you raise a great point about whether prior knowledge of span dependence (e.g., positional proximity) can be leveraged. A crude idea to leverage positional proximity would be to divide the sequence into larger spans, which reduces $m$ and therefore the chance that the supporting facts are incomplete after SpanDrop, as we have discussed in the response above. More task-dependent and/or uneven, discontiguous span segmentation techniques might prove beneficial as well, depending on the nature of the task. For instance, in some of our experiments we use sentences as spans (uneven lengths); in image tasks one might consider grouping 2-D neighbors together into a span even if they are not contiguous in the linearized sequence. Our derivations in Remarks 1 & 2 apply as long as these "spans" or "components" are disjoint.
> > >
> > > R2': While we do not have direct experimental results in this paper, we believe that SpanDrop can be equivalent to LengthDrop when applied on intermediate representations of neural networks with a span size of one, as we have mentioned in the response above. Therefore, the LengthDrop work provides sufficient evidence that a special case of (Bernoulli-)SpanDrop can be applied to intermediate representations and lead to performance gain. Since our focus and motivation of this paper is on creating counterfactual examples at the input level that is agnostic of specific model architectures, we leave further investigation of applying SpanDrop techniques to intermediate representations to future work.
> > >
> > > R3': In our experiments on the synthetic data, SpanDrop can sometimes lead to significant performance gains at the cost of 2-3x the number of training steps to converge (where the model trained without SpanDrop saturates). Note that the number of training steps is dependent on the task and the strength of regularization (the choice of drop rate $p$, and whether Beta-SpanDrop is used), and thus there is not a general rule of thumb. We will include further analysis in the revision of the paper.

---

> > > > ### Comment · Reviewer_EEvq · 2021-11-30
> > > > **Thanks for the prompt response**
> > > >
> > > > Your response helped me understand better. Thank you for the detailed explanation.

---

### Author Response · Authors · 2021-11-12
**Thank you for your helpful feedback!**

We thank the reviewers for the helpful feedback. A common concern among some reviewers is the novelty of our work, especially its relation to previous techniques like word dropout. To clarify the contribution we make in this paper:

1. To the best of our knowledge, “word dropout” is commonly used in the literature to refer to one of two related but different operations: a) “word masking”, where random words in the input are replaced with an artificial <unk> or <mask> tokens before fed into the model, and b) “word removal”, which usually removes intermediate model representations that correspond to random words before aggregating the result for sequence-level tasks. Word masking introduces artificial tokens that are only used at training time, which exposes the model to unnatural input sequences and distribution drift. On the other hand, prior word removal approaches typically require knowledge of the model architecture to know where sequence-level feature aggregation is performed.

2. In contrast, SpanDrop directly operates at the input level and makes no change to the underlying model, which can be treated as a black box. Moreover, we expand the object of consideration beyond single tokens or values in the input sequence, and instead introduce potentially larger units that are more meaningful given the properties of different tasks. As a result, the model is trained on counterfactual examples without artifacts like order perturbation or artificial symbols that are not present at test time. We also demonstrate in our synthetic data experiments that when applied at the word level, SpanDrop is more effective than word masking (SpanMask in the paper) at generalizing to unseen data, especially in mitigating position bias. Our experiments on real-world NLP datasets also demonstrate that SpanDrop can improve model performance regardless of the choice of span size (token, sentence, or a few consecutive tokens).

3. We also formulate the problem of long sequence inference with sparse supporting facts, and provide theoretical analysis of different properties of these techniques that are validated with our experiments. That is, we explicitly consider a setting that is different from what is commonly discussed in previous work, where the main goal is to construct models and benchmarks with sequence length $n$ as the only variable, with the implicit assumption that all $n$ variables should be considered to determine the output. For instance, in Tay et al.’s (2020) Long Range Arena, the ListOps task requires models to read a list of numbers and symbolic operations and derive the correct answer after simulating the calculation. Specifically, in the following example from this task, removing any part of the input could perceivably alter the output, and therefore it does not satisfy our $m \ll n$ assumption:

    > INPUT: [MAX 4 3 [MIN 2 3 ] 1 0 [MEDIAN 1 5 8 9, 2]] OUTPUT: 5

    On the modeling side, since SpanDrop can generate counterfactual examples to automatically ablate input variables when the true set of supporting facts is small ($m\ll n$), it is therefore orthogonal and complementary to long-sequence neural models, which are designed to process longer sequences with better computational and memory efficiency, when the kind of tasks we discuss are concerned.

4. Furthermore, based on our formulation and theoretical analysis, we also propose a novel variant of SpanDrop technique based on the Beta-Bernoulli distribution. We further validate the properties and contribution of Beta-SpanDrop on both synthetic data and real-world data. We also believe that our analysis regarding the effect of the Beta-Bernoulli distribution is applicable beyond the understanding of different SpanDrop variants, and can be applied to other forms of data augmentation or regularization techniques.

---

### Decision · Program_Chairs · 2022-01-20

**Decision:**

Reject

**Comment:**

The paper proposes a data augmentation approach called SpanDrop to help to distill supervision signals from a long sequence prediction problem. The reviewers generally agree on two major drawbacks of the paper. First, the novelty of this approach. Second, the experiment results are not very convincing.

After reading the responses from the authors, I don’t think the authors convinced me of the novelty of the work, especially when comparing it to the word dropout. No matter if you treat the data or the model as a black-box, it’s effectively doing the same thing. Apart from that, the model can only be used in the setting of “underspecification” long sequence tasks, which diminishes its value in real applications.

On the experiment side, there are three issues. One, many tasks considered are not long sequence tasks. Second, the improvement is marginal in many cases. Three, more related methods should get considered as baselines. Besides these three points raised by the reviewers, I also want to raise the point that it is not (and should not) be acceptable to report ALL your language experiment results on dev sets. I understand it is more time-consuming to get the test results on tasks where the test has to be done online, e.g. SQuAD. However, it is not a good practice in reaching a conclusion merely on dev sets in general.

Based on the reviewers' comments and the reasons listed above, I recommend rejection of this paper.